# Hunger and Satiety Peptides: Is There a Pattern to Classify Patients with Prader-Willi Syndrome?

**DOI:** 10.3390/jcm10215170

**Published:** 2021-11-04

**Authors:** Marta Bueno, Ester Boixadera-Planas, Laura Blanco-Hinojo, Susanna Esteba-Castillo, Olga Giménez-Palop, David Torrents-Rodas, Jesús Pujol, Raquel Corripio, Joan Deus, Assumpta Caixàs

**Affiliations:** 1Endocrinology and Nutrition Department, University Hospital Arnau de Vilanova, Obesity, Diabetes and Metabolism (ODIM) Research Group, IRBLleida, University of Lleida, 25198 Lleida, Spain; mbuenodiez@gmail.com; 2Servei d’Estadística Aplicada, Universitat Autònoma de Barcelona, 08193 Cerdanyola, Spain; ester.boixadera@uab.cat; 3MRI Research Unit, Department of Radiology, Hospital del Mar, 08003 Barcelona, Spain; laura.blanco02@gmail.com (L.B.-H.); 21404jpn@comb.cat (J.P.); joan.deus@uab.cat (J.D.); 4Centro Investigación Biomédica en Red de Salud Mental, CIBERSAM G21, 08003 Barcelona, Spain; 5Specialized Service in Mental Health and Intellectual Disability, Institute of Health Assistance (IAS), Parc Hospitalari Martí i Julià, 17190 Girona, Spain; susanna.esteba@ias.cat; 6Neurodevelopment Group [Girona Biomedical Research Institute]-IDIBGI, Institute of Health Assistance (IAS), Parc Hospitalari Martí i Julià, 17190 Girona, Spain; 7Endocrinology and Nutrition Department, Parc Taulí Hospital Universitari, Institut d’Investigació i Innovació Parc Taulí I3PT, Medicine Department, Universitat Autònoma de Barcelona, 08208 Sabadell, Spain; ogimenez@tauli.cat; 8Biological Psychology Department, Philipps-Universität Marburg, 35037 Marburg, Germany; torrents@staff.uni-marburg.de; 9Pediatric Endocrine Department, Parc Taulí Hospital Universitari, Institut d’Investigació i Innovació Parc Taulí I3PT, Medicine Department, Universitat Autònoma de Barcelona, 08208 Sabadell, Spain; rcorripio@tauli.cat; 10Department of Clinical and Health Psychology, Universitat Autònoma de Barcelona, 08193 Cerdanyola, Spain

**Keywords:** Prader-Willi syndrome, obesity, hunger, satiety, BDNF, PYY, ghrelin, clusters

## Abstract

Hyperphagia is one of the main problems of patients with Prader-Willi syndrome (PWS) to cope with everyday life. The underlying mechanisms are not yet well understood. Gut-brain hormones are an interrelated network that may be at least partially involved. We aimed to study the hormonal profile of PWS patients in comparison with obese and healthy controls. Thirty adult PWS patients (15 men; age 27.5 ± 8.02 years; BMI 32.4 ± 8.14 kg/m^2^), 30 obese and 30 healthy controls were studied before and after eating a hypercaloric liquid diet. Plasma brain-derived neurotrophic factor (BDNF), leptin, total and active ghrelin, peptide YY (PYY), pancreatic polypeptide (PP), Glucagon-like peptide-1 (GLP-1), glucose-dependent insulinotropic polypeptide (GIP) and amylin were determined at times 0′, 30′, 60′ and 120′. Cluster analysis was used. When considering all peptides together, two clusters were established according to fasting hormonal standardized concentrations. Cluster 1 encompassed most of obese (25/30) and healthy controls (28/30). By contrast, the majority of patients with PWS were located in Cluster 2 (23/27) and presented a similar fasting profile with hyperghrelinemia, high levels of leptin, PYY, GIP and GLP-1, compared to Cluster 1; that may reflect a dysfunction of these hunger/satiety hormones. When peptide behavior over the time was considered, PP concentrations were not sustained postprandially from 60 min onwards in Cluster 2. BDNF and amylin did not help to differentiate the two clusters. Thus, cluster analysis could be a good tool to distinguish and characterize the differences in hormone responses between PWS and obese or healthy controls.

## 1. Introduction

Prader-Willi syndrome (PWS) is the first genetic syndromic cause of obesity and is caused by the lack of expression of the genes contained in the 15q11-q13 region of chromosome 15 of paternal origin. The syndrome includes features such as neonatal hypotonia, intellectual disability, growth hormone deficiency or hypogonadism. Undoubtedly, its main characteristic is hyperphagia, which leads to the development of morbid obesity, the main cause of morbidity and mortality in these patients [1]. Knowing the underlying causes behind hyperphagia would enable the development of new therapeutic targets. However, although many studies have tried to delve into them, they are not yet well known.

In this sense, there are several works that have tried to find alterations in different hormones involved in the regulation of appetite. Thus, hyperghrelinemia is a constant finding in these patients [2,3,4,5]. They have high total ghrelin concentrations with high levels of active or acylated ghrelin and normal levels of unacylated ghrelin, reflecting a relative unacylated ghrelin deficit [6]. Most studies also agree finding low concentrations of pancreatic polypeptide (PP) [7,8,9]. The results are controversial with regard to other hormones such as peptide YY (PYY) [10,11,12,13,14,15] or Glucagon-like peptide-1 (GLP-1) [16,17,18,19], and there are few studies exploring others such as glucose-dependent insulinotropic polypeptide (GIP) [8,15], amylin [20] or brain-derived neurotrophic factor (BDNF) [21,22].

Given that ghrelin is a potent orexigen [23], the constant finding of high concentrations of such peptide in subjects with PWS suggests that this may be the cause of its hyperphagia. However, some studies show that hyperghrelinemia is present in very young children before the onset of hyperphagia [5,24] and the reduction of ghrelin concentrations with an infusion of somatostatin or with analogues thereof does not induce a reduction in appetite or body weight or a change in eating behavior in these patients [25,26]. Livoletide, an unacylated ghrelin analogue, showed initially a significant reduction in hyperphagia, waist circumference and fat mass in PWS subjects, without changes in ghrelin concentrations [27], but these effects were not sustained after three months of treatment [28]. PP is a peptide that induces satiety [29]. The finding of low concentrations in patients with PWS suggests that it may be involved in their lack of satiety. However, the effects of its infusion are controversial. In children at high doses, it does not decrease food intake [9]. With more physiological and repeated doses, a reduction in intake is achieved in adult women with PWS [30]. Treatment with GLP-1 receptor agonists is used in subjects with PWS. There is a study in rats [31] in which exendin-4 induces satiety and decreases ghrelin by 74%, and a report of a female PWS patient [32] in which liraglutide increases insulin and reduces GLP-1 and ghrelin concentrations. However, in other studies in rats [33] and humans [34,35], treatment with GLP-1 receptor agonists did not induce changes in ghrelin concentrations. No infusion or blocking experiments have been performed on BDNF, amylin or GIP in PWS or PWS animal models.

Treating children with PWS using brain peptides such as oxytocin, for a short period of time, resulted encouraging. A double-blind, placebo-controlled, crossover study with intranasal oxytocin (low doses) for five days or intranasal placebo spray, was safe and showed a reduction in appetite drive, and improvements in socialization, anxiety, and repetitive behaviors [36]. However, other studies for a longer period of time found no effect of oxytocin at 8 weeks of treatment [37] or a different degree of response at 3 months of treatment depending on sex and genetic subtype (boys and deletion were responders) [38].

Thus, appetite regulation is an extraordinarily complex process involving multiple peptides that interact with each other in a synergistic or antagonistic way. Therefore, it is plausible that hyperphagia in PWS is due to a failure of some overlapped peptide signals rather than to a single peptide alteration.

In the present study, we aimed to evaluate different peptides involved in the regulation of appetite in the PWS individuals and assess the possible existence of a hormonal pattern that may help to explain the exaggerated hyperphagia in this syndrome. We compared the hormonal pattern to age and sex matched obese and lean controls.

## 2. Materials and Methods

### 2.1. Participants

We studied 30 adult patients with genetically confirmed PWS, 30 obese controls matched by age, sex and body mass index (BMI) and 30 lean controls matched by age and sex. Genetic subtype was type 1 deletion (*n* = 7), type 2 deletion (*n* = 13), uniparental disomy (*n* = 7) and imprinting defects (*n* = 3). Patients with PWS and obese controls were recruited from the Endocrinology and Nutrition Department and lean controls were hospital staff or acquaintances volunteers. The study was approved by the local ethics committee (Comitè d’Ètica d’Investigació amb medicaments del Parc Taulí Ref 2011/511) in January 2011. The informed consent of the participants or their parents or guardians was obtained, and the investigations were carried out in accordance with the Helsinki Declaration.

All subjects had a stable weight for at least three months before the inclusion. No patient with PWS was receiving GH treatment at the time of the study, although 14 had previously received it. Twenty-seven had hypogonadism, of which 10 were in treatment. Seven PWS patients and one obese control had type 2 diabetes, being treated with metformin alone or in combination with GLP1 receptor agonists, DPP4 inhibitors or insulin.

### 2.2. Experimental Methods

Subjects were admitted at 8 a.m. after an overnight fast of at least 10 hours. Anthropometric measures were taken, and body composition analysis was performed using bioelectrical impedance (TANITA, body composition analyzer BC-418 MA Biologica Tecnologia Medica SL-BCN, Tokyo, Japan). A blood sample was taken for routine analysis and fasting hormonal analysis.

For the postprandial study, subjects ingested a standard liquid diet (Resource 2.0, Nestle Lab, 1200 kcal, 43% carbohydrate, 39% fat, 18% protein) and blood samples were drawn at 30, 60 and 120 min after the intake. All samples were collected on ice and spun at 4 °C. Plasma was centrifuged and stored at −80 °C until processed.

Assays were performed using commercially available methods. BDNF was quantified by an enzyme-linked immunosorbent assay (ELISA) kit (Cat. No. CYT306, Millipore; Billerica, MA, USA) following the manufacturers’ instructions. This kit’s intra-assay variation is ±3.7% and its inter-assay variation is ±8.5% (125 pg/mL). Ghrelin (total and active), amylin, GIP, GLP-1, insulin, leptin, PP and PYY concentrations were determined using Luminex 200 and the human metabolic hormone magnetic bead panel (HMHMAG-34K, Merck Millipore; Billerica, MA, USA) following the manufacturer’s instructions. The sensitivity of this kit is as follows: total and active ghrelin 11 pg/mL, amylin 4 pg/mL, GIP 0.7 pg/mL, GLP-1 20 pg/mL, insulin 48 pg/mL, leptin 184 pg/mL, PP 5 pg/mL, PYY 26 pg/mL. The intra and inter assay coefficients of variation are, respectively as follows: total and active ghrelin 3% and 21%, amylin 3% and 7%, GIP 4% and 10%, GLP-1 2% and 13%, insulin 3% and 10%, leptin 7% and 10%, PP 6% and 9%, PYY 7% and 8%.

Subjects (three patients with PWS) receiving GLP1 receptor agonists or dipeptidil peptidasa-4 (DPP4) inhibitors for diabetes treatment were excluded for GLP-1 and GIP determinations and for cluster analysis.

Before the meal and 60′ and 120′ after the meal, subjects quantified their hunger on a visual analogue scale ranging from 0 to 100.

### 2.3. Statistical Methods

We used a Kruskal-Wallis test and a Mann-Whitney test with Bonferroni correction for pairwise comparisons between groups. We used Wilcoxon signed rank test to detect statistically significant differences of the concentration evolution (from baseline).

We elaborated a generalized linear regression model with repeated measures for an indicator of values higher than the median score for each peptide including cluster_t0 and time and the interaction between them.

A hierarchical cluster analysis [39] was conducted to reveal groups of patients with similar hormonal concentrations based on the factors derived from the principal components analysis. The clusters obtained are represented in a tree dendrogram. The distribution of the sample within each cluster was analyzed using Mann-Whitney-Wilcoxon test for continuous variables and chi-square test for categorical variables.

A classification tree analysis [40] was conducted to discriminate patient groups (PWS, obese, lean controls) through hormonal concentrations levels. Entropy and the cost-complexity criterion were used to build the tree. The number of terminal nodes was determined using 10-fold cross validation.

We used SAS system v9.4 (SAS Institute Inc., Cary, NC, USA) for all analyses.

## 3. Results

Table 1 shows subjects’ baseline characteristics. Lean controls had a lower BMI, percentage of body fat and waist circumference than PWS subjects and obese controls. Five subjects on insulin treatment were excluded for insulin determinations. Fasting glucose levels were lower in lean subjects than in obese controls. Insulin and insulin resistance determined by homeostatic model assessment (HOMA-IR-index) were higher in obese controls than in PWS subjects and lean controls.

### 3.1. Fasting Study

Table 2 shows the results of the fasting study.

Total ghrelin concentrations were higher in PWS patients than in obese controls (stat = 604.00, *p* < 0.001). Active ghrelin tended to higher values in PWS subjects than in controls. Plasma BDNF levels tended to be lower (χ^2^ = 5.78, *p* = 0.05) in PWS patients than in obese and lean controls. Leptin concentrations were higher in PWS subjects than in controls and lower in lean than in obese controls (χ^2^ = 55.7, *p* < 0.001). Fasting PYY was higher in PWS than in controls and higher in obese than in lean subjects (χ^2^ = 49.05, *p* < 0.001). GLP-1 concentrations were higher in PWS (χ^2^ = 30.91, *p* < 0.001). No differences were found in fasting concentrations of GIP, amylin or PP. Statistically significant pairwise comparisons are indicated in Table 2.

### 3.2. Postprandial Study

The 60′ sample could not be collected in one subject with PWS.

Patients with PWS presented higher concentrations of leptin than both control groups (*p* < 0.001), with no changes in leptin concentrations in PWS, and a slight decrease in obese and lean controls (Figure 1, leptin graph was already published elsewhere [22]).

Total ghrelin decreased in all three groups after ingestion (*p* < 0.001), although its concentrations were always higher (*p* < 0.001) in subjects with PWS than in control groups. Active ghrelin concentrations tended to be higher in subjects with PWS than in the two control groups in the first 30 min after ingestion and decreased significantly after 60 min in all groups (*p* < 0.02) (Figure 1).

In lean controls, a peak of BDNF occurred at 60′ after ingestion (S = 114.5, *p* = 0.016) while in PWS was earlier, truncated and lower at one hour after ingestion (S = 102.5, *p* = 0.024) (Figure 1, BDNF graph was already published elsewhere [22]).

Obese and healthy controls showed an increase in PP after ingestion, with a peak at 30 min (*p* < 0.001). The peak of PP in PWS subjects was truncated, with the lowest peak values for this group (Figure 1).

PYY and GLP-1 concentrations increased postprandially in PWS subjects (*p* < 0.001 and *p* < 0.001), who presented higher concentrations of both peptides than control groups (*p* < 0.001 at all times) (Figure 1).

GIP increased after ingestion in all three groups (*p* < 0.001). Postprandial concentrations of GIP were lower in lean controls than in subjects with PWS at all times and lower in lean controls than in obese controls at 30 and 60 min after ingestion (*p* < 0.01) (Figure 1).

Postprandial amylin increased in all three groups (*p* < 0.001). No significant differences among groups were found until t = 120, being the concentration at that time higher in PWS (Figure 1).

### 3.3. Cluster Analysis

Clusters were grouped considering the standardized concentrations of the nine peptides studied in three different times, time 0 (baseline), time 60 (postprandial), and all peptides over the time, so subjects who were within the same cluster presented more similarities than those in the other cluster. For cluster analysis, only subjects with complete data were considered (27 subjects with PWS, 30 with obesity and 30 heathy).

Considering baseline peptides (time 0), Cluster 1 included the majority (28/30) of healthy controls and (25/30) obese controls and only four subjects with PWS, while Cluster 2 includes the majority (23/27) of patients with PWS but only five obese controls and two healthy controls (Figure 2).

When clusters were grouped considering the nine peptides postprandially at time 60, patients with PWS were mostly placed in the same cluster as controls (16/26 subjects with PWS, 26/30 obese controls and 30/30 healthy controls in Cluster 1). Thus, this analysis did not differentiate PWS subjects from controls.

When we defined clusters globally (considering all peptides over the time), the situation was similar to time 60 (PWS widely distributed between the two groups).

Since clusters grouped by peptides at baseline discriminated better PWS from controls, we analyzed the clusters defined in this way (Figure 2).

Subjects placed in Cluster 2 (mainly PWS subjects) showed higher BMI (32.19 ± 8.15 vs. 27.88 ± 7.96 kg/m2, stat = 1629, *p* = 0.007), waist circumference (103.47 ± 18.47 vs. 92.37 ± 18.97 cm, stat = 1618.5, *p* = 0.009), and % of body fat (35.89 ± 9.92 vs. 28.36 ± 11.45%, stat = 1663.5, *p* = 0.003) than Cluster 1 subjects. No differences between clusters were observed in gender, presence of type 2 DM or HOMA-IR. Fasting hunger was similar in both clusters but subjects in Cluster 2 had more postprandial hunger (31.83 ± 36.64 vs. 8.95 ± 15.63, stat = 1633.5, *p* = 0.003). There were no differences in fasting glucose or insulin concentrations between clusters but subjects in Cluster 2 showed higher postprandial insulin concentrations (3449.61 ± 2309.52 vs. 2011.04 ± 1159.78 μU/mL, stat = 1551, *p* = 0.002 at 30′).

Subjects in Cluster 2 showed higher fasting concentrations of GIP (*p* < 0.002), GLP-1 (*p* < 0.0001), PYY (*p* < 0.0001), leptin (*p* < 0.0001) and active ghrelin (*p* < 0.04), and higher postprandial concentrations of total and active ghrelin, GIP, GLP-1, PYY and leptin (*p* < 0.05) than those in Cluster 1. There were no differences in BDNF, PP or amylin concentrations between the two clusters (Figure 3).

We examined peptide behavior over the time in Cluster 1 and Cluster 2 using a logistic regression model with repeated measures of the binary outcome “Concentration higher than median score”. The only statistically significant interaction between cluster and time was PP peptide (F = 3.36, *p* = 0.02). As we can see in the Figure 4, the percentage of participants with a “concentration higher than median of PP peptide” in Cluster 1 increased after ingestion in both clusters until time 60′ (*p* < 0.001), to be maintained onwards in Cluster 1 but not in Cluster 2 (Figure 4). With respect to the principal effects models (with no interaction), there was a time effect in some peptides: amylin, GIP and PYY increased over time (*p* < 0.001); active and total ghrelin decreased (*p* < 0.003) over time; and BDNF, GLP-1 and leptin did not change significantly above time.

The 23 patients with PWS included in Cluster 2 had lower BMI (31.79 ± 7.97 vs. 37.28 ± 11.6 kg/m^2^), lower waist circumference (104.09 ± 17.94 vs. 116.88 ± 24.85 cm) and % of body fat (36.67 ± 7.48 vs. 42.3 ± 12.62) than those in Cluster 1. Regarding the genetic subtype, patients located in Cluster 2 had type 1 deletion (5), type 2 deletion (9), uniparental disomy (6) or imprinting defects (3). The four PWS located in Cluster 1 had type 1 deletion (1), type 2 deletion (2) or uniparental disomy (1).

### 3.4. Classification Tree

We also built a classification tree (Figure 5) in order to discriminate which peptide was most important to define the three groups of subjects. We considered all possible cutoff of fasting peptide concentrations; the sample was split for the best discrimination level of groups. This process was repeated on each derived subset in a recursive manner called recursive partitioning. After this process, we decided the number of leaves using 10-fold cross-validation:Leave 1: Leptin < 15.3 ng/mL;Leave 3: Leptin ≥ 15.3 ng/mL and GLP-1 < 52.5 pg/mL;Leave 4: Leptin ≥ 15.3 ng/mL and GLP-1 ≥ 52.5 pg/mL.

We found that plasma leptin concentrations were the best to distinguish healthy subjects from the other groups (leptin was lower in healthy subjects). Afterwards, GLP-1 was the principal peptide to discriminate between obese subjects with and without PWS (GLP-1 was higher in PWS subjects).

Table 3 shows the distribution of subject groups across final leaves of the classification tree.

## 4. Discussion

Prader-Willi syndrome (PWS) is a rare neurodevelopmental disorder associated with intellectual disability, hypotonia, hyperphagia (an insatiable hunger), and obesity. Mechanisms underlying hyperphagia are not yet well understood. Appetite regulation is an extraordinarily complex process. Gut-brain hormones are an interrelated network that may be at least partially involved. Here, we propose cluster analysis to differentiate patients with PWS from obese and healthy controls through their hormonal pattern.

In the present study, clusters were grouped considering three times, fasting, postprandial and all peptides over the time. The main finding of this study is that fasting peptides classify better patients with PWS than postprandial peptides. In that sense, two clusters were obtained considering fasting peptides. Cluster 1 grouped mainly healthy and obese control subjects. Cluster 2 grouped mainly patients with PWS. The characteristic hormonal pattern of Cluster 2 consisted of hyperghrelinemia (both total and active), high levels of leptin, PYY, GIP and GLP-1. Amylin and BDNF did not differ between clusters. The only peptide that presented a different postprandial behavior between clusters was PP.

Trying to analyze thoroughly the hormonal pattern obtained, leptin was higher in Cluster 2 subjects (than in Cluster 1) according to their higher BMI and % of body fat, since the main source of leptin is white adipose tissue cells [41]. However, leptin is not a good differentiator between PWS and obese subjects, as observed in the classification tree.

One of Cluster 2 most relevant characteristics was the elevated concentrations of active and total ghrelin. This could explain the lack of satiety in these patients, since ghrelin is an orexigenic signal and its high fasting and postprandial concentrations in PWS are widely described in the literature [2,3,4,5]. In accordance, subjects in Cluster 2 showed more postprandial hunger than those in Cluster 1. However, these subjects also showed greater concentrations of GIP, GLP-1, and PYY; all of them anorexigenic signals. Haqq et al. [15] suggested that elevated satiety signal concentrations could be a compensatory response to higher ghrelin levels. Additionally, oversecretion of PYY or GIP could be a manifestation of disordered autonomic or vagal efferent input to the gastrointestinal tract. In this sense, although subjects with PWS (in Cluster 2) have higher concentrations of GLP-1, PYY or GIP than control subjects (in Cluster 1), they do not exert a sufficient satiety effect, as if there was a certain component of hormone resistance. This resistance could be higher for GLP-1 and PYY, which are the peptides that show more extreme values in subjects with PWS in our study. In fact, GLP-1 is the peptide which differentiates PWS from obese controls, in our classification tree.

GIP concentrations in PWS are only evaluated in children with discordant results [8,15,42]. Haqq et al. [15] described higher fasting GIP concentrations than controls, but Zipf et al. [8] described lower GIP concentrations. Some studies [16,17,18] did not observe differences either in fasting or in postprandial GLP-1 concentrations between PWS and control subjects, while in another study [19] there were postprandial differences in relation to the speed of food intake (more GLP-1 secretion with rapid consumption). We found higher concentrations of GLP-1, both fasting and postprandially in Cluster 2 compared to Cluster 1. Carbohydrate restriction is described to increase fasting and post-prandial GLP-1 concentrations in PWS children [42]. Unfortunately, we have no data about the caloric content and composition of the diet that the participants had followed prior to the study, but since subjects with PWS are usually urged to follow a low-calorie diet, it is possible that they also tended to restrict carbohydrate intake and this fact could contribute to their higher fasting GLP-1 concentrations. Higher postprandial GLP1 levels in Cluster 2 than in Cluster 1 could be justified by the speed of ingestion, which was very fast in patients with PWS due to the liquid consistency of the meal and their anxiety for food intake. The composition of the meal, specifically fat content (39% of fat, 468 Kcal), a stimulator of GLP-1 secretion may have influenced plasma levels, but all participants in both clusters ate the same caloric intake and fat content. Studies evaluating PYY concentrations in PWS show mixed results [10,11,12,13,14,15]. As with GLP-1, in children [19], the higher the speed in food intake, the higher PYY postprandial secretion. In this sense, the composition of food or the speed of consumption could also account for the present results and the discordances in the postprandial results found between studies.

We did not observe any differences in BDNF or amylin concentrations between the two clusters. To our knowledge, there are three studies in the literature about plasma BDNF in PWS. The first one [21] reports low fasting BDNF concentrations in PWS children compared to controls. The other two were performed in adult patients and observed similar results in different contexts. One reported no differences in plasma BDNF levels between participants with PWS and obese controls before and after exercising, although BDNF response in PWS was numerically lower than in controls [43]. The other one, published by our group [22] showed a tendency to lower fasting BDNF concentrations in PWS than in controls and a postprandial BDNF truncated peak that could contribute to the lack of satiety (data also shown in Figure 1 of this manuscript). However, in the present study, cluster analysis did not show differences in BDNF between Cluster 1 (obese and healthy) and Cluster 2 (mainly PWS), suggesting that BDNF may not be the principal hormone signal to drive the lack of satiety in PWS. However, since the lack of BDNF postprandial peak was also shown in obese patients [22], BDNF postprandial peak in healthy subjects may have been masked by the lack of peak in obese (both in Cluster 1) and contributed to the absence of differences between clusters. There is only one study [20] evaluating amylin in children with PWS finding lower postprandial amylin concentrations which correlated with insulin concentrations. A relative hypoinsulinemia is described in PWS [16,17,44], and fasting insulin levels in PWS in the present study were in the same line, however, no differences between clusters were observed. After the meal, insulin concentrations increased in both clusters but were always higher in Cluster 2, despite the same proportion of type 2 diabetes and the exclusion of subjects on insulin treatment for insulin analysis.

We found no differences in fasting PP concentrations between clusters, although most studies report lower concentrations in PWS subjects [7,8,9]. However, postprandial PP increase in concentration failed to be maintained only in Cluster 2. Although the truncated postprandial PP response was previously described [7,8], it was the only peptide that behaved differently postprandially between clusters in the present study and may help to explain the lack of satiety in this syndrome.

Given that cluster analysis grouped the majority of participants with PWS in Cluster 2, we could speculate that those that fell into Cluster 1 were more similar to controls. However, to our surprise, PWS of Cluster 2 had lower BMI, waist circumference and % body fat than those of Cluster 1 (statistical analysis not performed due to the small number of PWS in Cluster 1). Thus, a limitation of the present study is the small number of participants included that do not allow certain calculations, in spite of being an acceptable number for a rare disease. Another limitation would be the inclusion of only 9 hunger/satiety signals, although we chose the most usually studied in the literature. The strength of the study is the use of both fasting and postprandial peptide concentrations. Since patients with PWS present higher postprandial hunger than controls [22], one could hypothesize that postprandial peptides could differentiate better PWS patients from controls. In the present study, we have demonstrated that fasting peptides are better definers of hunger/satiety hormone profile in PWS than postprandial peptides, since all except PP present the same postprandial behavior in both clusters.

In summary, the authors have used cluster analysis to obtain a hormone pattern to classify patients with PWS. The majority of them were grouped in Cluster 2 and presented a similar fasting profile with hyperghrelinemia, high levels of leptin, PYY, GIP and GLP-1, compared to Cluster 1 (majority of controls), that may reflect a dysfunction of these hunger/satiety hormones. When peptide behavior over the time was considered, PP concentrations were not sustained postprandially in Cluster 2. BDNF and amylin did not help to differentiate the two clusters. Thus, cluster analysis could be a good tool to distinguish and characterize the differences in hormone responses between PWS and obese or healthy controls. The use of clusters may be of help to elucidate which peptides should be targeted for future drug development.

## Figures and Tables

**Figure 1 jcm-10-05170-f001:**
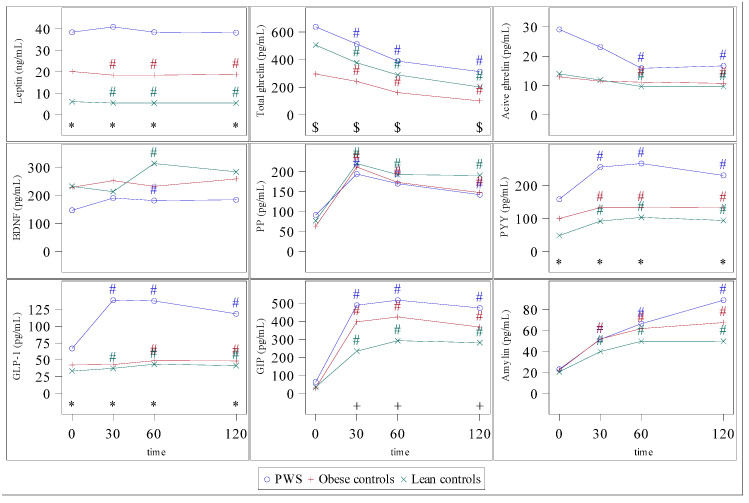
Postprandial hormonal concentrations. # *p* < 0.05 vs. basal concentrations. * *p* < 0.05 between groups. $ *p* < 0.05 PWS vs. obese controls. + *p* < 0.005 PWS vs. lean controls. Plasma brain-derived neurotrophic factor (BDNF), peptide YY (PYY), pancreatic polypeptide (PP), Glucagon-like peptide-1 (GLP-1), glucose-dependent insulinotropic polypeptide (GIP).

**Figure 2 jcm-10-05170-f002:**
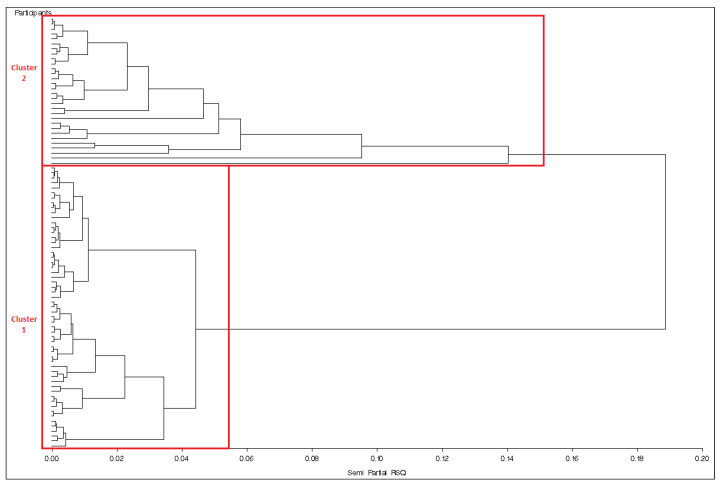
Clusters according to baseline peptides.

**Figure 3 jcm-10-05170-f003:**
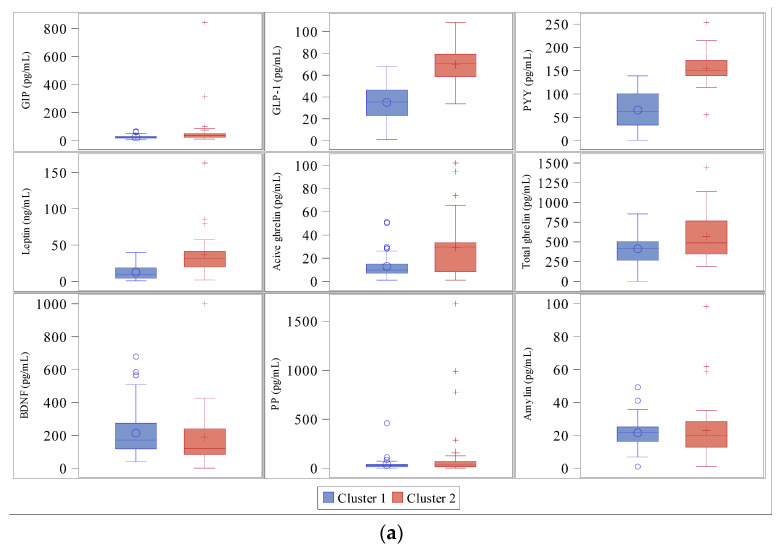
Hormonal concentrations between clusters. Glucose-dependent insulinotropic polypeptide (GIP), Glucagon-like peptide-1 (GLP-1), peptide YY (PYY), plasma brain-derived neurotrophic factor (BDNF), pancreatic polypeptide (PP). (**a**) At fasting. GIP (*p* < 0.002), GLP-1 (*p* < 0.0001), PYY (*p* < 0.0001), leptin (*p* < 0.0001) and active ghrelin (*p* < 0.04); (**b**) At time 60′. Total and active ghrelin, GIP, GLP-1, PYY and leptin (*p* < 0.05). Cluster 1 outliers (o) and cluster 2 outliers (+).

**Figure 4 jcm-10-05170-f004:**
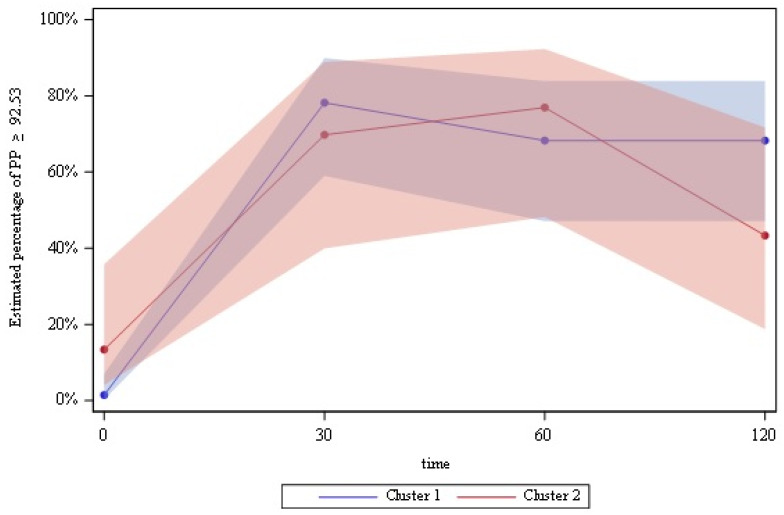
Postprandial PP concentrations in baseline clusters.

**Figure 5 jcm-10-05170-f005:**
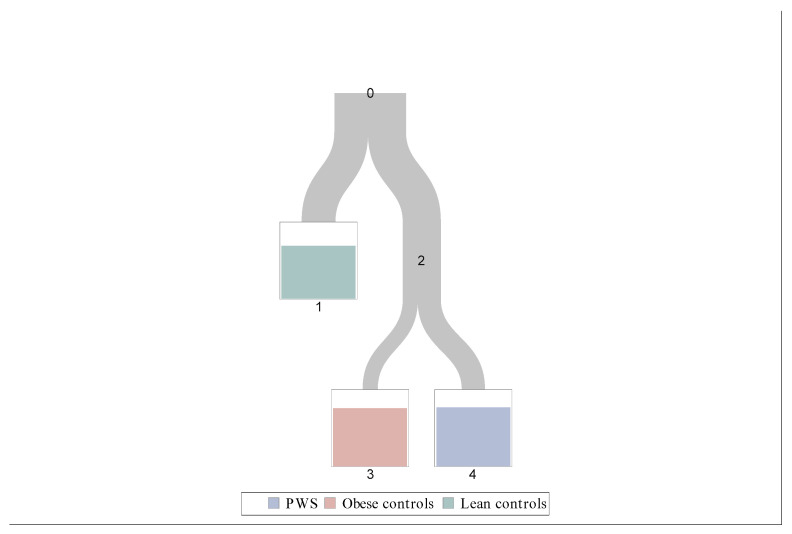
Classification tree according to fasting peptide concentrations. 0 All participants, 1 mostly leans controls, 2 mostly obese and Prader-Willi (PWS) subjects, 3 mostly obese subjects, 4 mostly PWS subjects.

**Table 1 jcm-10-05170-t001:** Baseline characteristics of all groups.

	PWS (*n* = 30)	Obese Controls (*n* = 30)	Lean Controls (*n* = 30)
Sex (M/F)	15/15	15/15	15/15
Age (year)	27.5 ± 8.02	28.4 ± 7.13	27.9 ± 7.77
BMI (kg/m^2^)	32.4 ± 8.14	33.7 ± 6.88	22.1 ± 2.05 *^$^
Body fat (%)	37.0 ± 8.39	35.7 ± 9.70	20.3 ± 7.23 *^$^
Waist (cm)	105.0 ± 18.3	105.3 ± 15.1	78.3 ± 7.43 *^$^
Glucose (mg/dL)	94.1 ± 22.0	98.2 ± 32.1	86.7 ± 6.64 ^$^
Insulin (μU/mL)	333.7 ± 218.5	493.2 ± 213.7 *	304.8 ± 134.7 ^$^
HOMA-IR	1.69 ± 0.84	2.94 ± 1.57 *	1.69 ± 0.77 ^$^

*p* < 0.05 vs. PWS * and obese ^$^ patients. All quantitative values are expressed as mean ± SD. PWS: Prader-Willi Syndrome; BMI: body mass index; HOMA-IR: homeostatic model assessment insulin resistance.

**Table 2 jcm-10-05170-t002:** Fasting hormonal concentrations.

	PWS	Obese Controls	Lean Controls
BDNF (pg/mL)	113.7 (75.6–237.7)	187.6 (119.3–303.0)	158.2 (118.0–275.8)
Leptin (ng/mL)	35.3 (20.8–42.9)	18.5 (12.8–25.4) *	4.4 (2.7–9.2) *^$^
Total ghrelin (pg/mL)	572.4 (379.8–812.5)	279.0 (199.4–396.2) *	477.2 (350.7–633.6) ^$^
Active ghrelin (pg/mL)	29.0 (1.0–35.8)	9.5 (5.5–15.0)	12.3 (8.6–16.0)
PYY (pg/mL)	149.2 (134.2–176.6)	101.3 (72.0–134.3) *	34.6 (1.0–67.7) *^$^
PP (pg/mL)	29.9 (9.5–55.3)	23.5 (12.5–38.3)	31.8 (23.7–61.2)
GLP-1 (pg/mL)	68.5 (55.8–77.0)	38.1 (33.7–53.3) *	32.5 (19.3–49.6) *
GIP (pg/mL)	30.2 (21.1–47.4)	27.5 (18.5–35.6)	24.8 (19.6–32.7)
Amylin (pg/mL)	19.9 (7.8–28.5)	22.4 (15.6–27.1)	20.9 (16.2–24.2)

*p* < 0.05 vs. PWS * and obese ^$^ patients. All quantitative values are expressed as median (interquartile range). Plasma brain-derived neurotrophic factor (BDNF), peptide YY (PYY), pancreatic polypeptide (PP), Glucagon-like peptide-1 (GLP-1), glucose-dependent insulinotropic polypeptide (GIP).

**Table 3 jcm-10-05170-t003:** Contingency table of Classification Tree Leaves by subject group.

Classification Tree Leaves	PWS	Obese Controls	Lean Controls
Leave 1	2 (4.9%)	10 (24.4%)	29 (**70.7%**)
Leave 3	3 (16.7%)	14 (**77.8%**)	1 (5.6%)
Leave 4	22 (**77.8%**)	6 (21.4%)	0 (0%)

Count and row percentage are reported. Percentages in bold indicate the main type of subjects in each leave. When we built a classification tree considering postprandial peptide concentrations, GLP-1 and leptin were also the main discriminant peptides between groups (data not shown).

## Data Availability

Data supporting reported results can be found in Statistics Department, Universitat Autònoma de Barcelona or corresponding author acaixas@tauli.cat).

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
