# Peer review of "Hunger and Satiety Peptides: Is There a Pattern to Classify Patients with Prader-Willi Syndrome?"

_jcm, 2021, doi:10.3390/jcm10215170_

Round 1

Reviewer 1 Report

Summary

The authors studied various orexigenic and anorexigenic hormones in adults with Prader-Willi, obesity and lean controls.  They studied levels before and after a meal and did cluster analysis to determine factors associated with Prader Willi.  They found the best discrimination with fasting values and that Prader Willi subjects were primarily in Cluster 2 with higher levels of ghrelin, leptin, PYY, GIP, and GLP1. They do a good job of reviewing the literature and explaining their data.  I have no concerns. 

Author Response

We appreciate the reviewer's comments.

Reviewer 2 Report

This mechanistic analysis of hunger-related peptides in patients with PWS relative to lean controls and control with obesity is an ambitious and important study. 

The review of literature in the introduction is clear and concise. I would like to recommend the authors consider also discussing findings related to oxytocin administration and hyperphagia/appetite in people with PWS (Miller et al., Oxytocin treatment in children with Prader-Willi syndrome: A double-blind, placebo-controlled, crossover study. Am J Med Genet A. 2017 May;173(5):1243-1250. doi: 10.1002/ajmg.a.38160. Epub 2017 Mar 30. PMID: 28371242; PMCID: PMC5828021). Oxytocin appears to be a potentially effective treatment for hyperphagia and discussing the literature and the findings of the authors in the context of what is known about oxytocin in PWS is relevant to evaluating this potential treatment and other potential treatments.

Author Response

As suggested by the reviewer we have added a paragraph about oxytocin treatment for hyperphagia and behavior in children with PWS and the corresponding references. It is highlighted in yellow.
